# Immortalization of Different Breast Epithelial Cell Types Results in Distinct Mitochondrial Mutagenesis

**DOI:** 10.3390/ijms20112813

**Published:** 2019-06-08

**Authors:** Sujin Kwon, Susan S. Kim, Howard E. Nebeck, Eun Hyun Ahn

**Affiliations:** 1Department of Pathology, University of Washington, Seattle, WA 98195, USA; 2Department of Biochemistry, University of Washington, Seattle, WA 98195, USA; 3Institute of Stem Cell and Regenerative Medicine, University of Washington, Seattle, WA 98109, USA

**Keywords:** mitochondrial DNA, rare mutation, stem cells, breast cancer, mitochondrial tRNA, duplex sequencing, next generation sequencing

## Abstract

Different phenotypes of normal cells might influence genetic profiles, epigenetic profiles, and tumorigenicities of their transformed derivatives. In this study, we investigate whether the whole mitochondrial genome of immortalized cells can be attributed to the different phenotypes (stem vs. non-stem) of their normal epithelial cell originators. To accurately determine mutations, we employed Duplex Sequencing, which exhibits the lowest error rates among currently-available DNA sequencing methods. Our results indicate that the vast majority of the observed mutations of the whole mitochondrial DNA occur at low-frequency (rare mutations). The most prevalent rare mutation types are C→T/G→A and A→G/T→C transitions. Frequencies and spectra of homoplasmic point mutations are virtually identical between stem cell-derived immortalized (SV1) cells and non-stem cell-derived immortalized (SV22) cells, verifying that both cell types were derived from the same woman. However, frequencies of rare point mutations are significantly lower in SV1 cells (5.79 × 10^−5^) than in SV22 cells (1.16 × 10^−4^). The significantly lower frequencies of rare mutations are aligned with a finding of longer average distances to adjacent mutations in SV1 cells than in SV22 cells. Additionally, the predicted pathogenicity for rare mutations in the mitochondrial tRNA genes tends to be lower (by 2.5-fold) in SV1 cells than in SV22 cells. While four known/confirmed pathogenic mt-tRNA mutations (m.5650 G>A, m.5521 G>A, m.5690 A>G, m.1630 A>G) were identified in SV22 cells, no such mutations were observed in SV1 cells. Our findings suggest that the immortalization of normal cells with stem cell features leads to decreased mitochondrial mutagenesis, particularly in RNA gene regions. The mutation spectra and mutations specific to stem cell-derived immortalized cells (vs. non-stem cell derived) have implications in characterizing the heterogeneity of tumors and understanding the role of mitochondrial mutations in the immortalization and transformation of human cells.

## 1. Introduction

Evidence exists that distinct phenotypes of normal cell [1,2,3,4,5] or precancerous cell [6] originators for tumor derivatives lead to tumor heterogeneities. For example, Ince et al. (2007) reported that breast tumorigenic cells transformed from two different normal epithelial cell types (BPECs; HMECs) exhibited marked differences in histopathology, tumorigenicity, and metastatic abilities. Transformed BPECs caused lung metastases, whereas transformed HMECs were nonmetastatic [4]. Transformed BPECs were up to 104-fold more tumorigenic than transformed HMECs. However, no study has investigated subclonal mutations for transformed derivatives of different normal cell types.

Here, to study consequences of normal cell origins on genetic changes in their immortalized derivatives, we examined two different immortalized (pre-neoplastic) breast epithelial cell types that were derived from normal human breast epithelial cells (HBECs) with different phenotypes (Types I and II). Both Type-I and Type-II normal HBECs were isolated from breast tissue of the same woman; however, they exhibited different phenotypes. Type-I HBECs display stem cell characteristics and have been characterized by: the expression of a stem cell marker octamer-binding transcription factor 4 (Oct-4) [7] and estrogen receptor (ER)-α [8], and luminal epithelial markers [9,10]; a deficiency in gap-junction associated intercellular communication (GJIC) [9,11]; the ability to display anchorage-independent growth [11]; the ability to differentiate into Type-II (normal differentiated) HBECs [9,11]; reduced expression of maspin [12]; and the ability to form budding/ductal organoids on Matrigel in conjunction with Type-II HBECs [11]. In contrast, Type-II HBECs exhibit opposite phenotypes in the described features above (i.e., do not express stem cell markers; do not express ER-α; express basal epithelial markers; express GJIC proteins; higher expression of maspin).

Both Type-I (stem cell features) and Type-II (without stem cell features) HBECs were transformed with Simian virus 40 large T-antigen (SV40-T) into immortal/non-tumorigenic (pre-neoplastic) cells, M13SV1 and M13SV22, respectively. SV40-T is widely used to immortalize and transform mammalian cells [13,14]. Type-II HBECs are less susceptible to SV40-T transformation than Type-I HBECs, and rarely become immortal following transfection with SV40-T [8,9,11,15]. This suggests that Type-I HBECs and Type-I HBEC-derived immortalized cells appear to be the major target cells for breast carcinogenesis and transformation.

Previously, we compared mutations of the whole mitochondrial genome of normal HBECs (Type-I and Type-II) derived from three independent women [16]. In the present study, we examined whether the whole mitochondrial genome of different immortalized human breast epithelial cell types is influenced by phenotypes of their originator normal HBECs (Type-I and Type-II). Somatic mutations in mitochondrial (mt) DNA have been reported to contribute to breast cancer metastasis and are believed to play a crucial role in the pathophysiology of mitochondrial disease [17,18]. Several studies have made associations between breast cancer metastasis and mtDNA mutations in protein-coding genes such as mt-ATP6, mt-ND5, and mt-ND3 [19,20,21]. Mitochondrial RNA genes are important in protein translation, and mutations in them can also have significant consequences in human breast cancer [21,22]. Thus, we investigated these distinct regions (13 protein-coding genes and 2 rRNAs- and 22 tRNAs-encoding genes) of the whole mitochondrial genome for both types of immortalized HBECs (M13SV1; M13SV22).

To accurately determine low-frequency (heteroplasmic) rare and subclonal mutations, as well as high-frequency (homoplasmic) mutations, we applied Duplex Sequencing (DS), which detects mutations with unprecedented accuracy [23,24,25,26]. Unlike conventional sequencing technologies that sequence only a single strand of DNA, DS sequences both strands of DNA and scores mutations only if they are present in both strands of the same DNA molecule as complementary substitutions. This significantly lowers error rates (eg., Next generation sequencing error rates 10^−2^ to 10^−3^; DS error rates < 5 × 10^−8^) [23,24,26,27,28].

## 2. Results

In this study, non-stem cell-derived immortalized human breast epithelial cells (HBECs) will be referred to as SV22, and stem cell-derived immortalized HBECs will be referred to as SV1 cells. Both SV22 and SV1 cells were cultured under the same conditions, DNA was extracted, and DNA libraries were prepared for Duplex Sequencing as we have described previously [16]. The average number of nucleotides at each genome position (depth) was calculated as the total number of duplex consensus sequence (DCS) nucleotides sequenced divided by the mtDNA size of 16,569 bases. The average DCS depths were 1060 (all Figures and Tables) and 446 (Appendix A) for SV22 cells and 1666 (all Figures and Tables) and 2738 (Appendix A) for SV1 cells. These are estimated to be equivalent to single-strand tag-based sequencing depths of approximately 5000 to 6200 and 7900 to 9800, respectively [26].

We have defined mtDNA mutations (variants) found in SV22 and SV1 cells as homoplasmic (90–100%), subclonal (0–10%), and rare (0–1%) based on the mutation occurrence (%) at each genome position. Maternally inherited mitochondrial mutations arising during early embryonic development are more likely to be homoplasmic [29,30]. In the present study, we focused on rare and subclonal variants, as they presumably represent *de novo* somatic variants. Rare or subclonal mutations are not accurately determined by conventional sequencing methods due to their high background error frequencies [27,28,31]. These rare and subclonal mutations, however, are accurately detectable by Duplex Sequencing [23,24,25,26].

### 2.1. Both SV22 and SV1 Cells Exhibit Identical Homoplasmic Mutations, Verifying that Both Cell Types were Derived from the Same Individual

Thirty-five identical homoplasmic unique mutations were found between the two cell types (Appendix A). Frequencies, types (%), and context fractions (%) of homoplasmic mutations were almost identical (Appendix A) in both cell types. T>C/A>G and C>T/G>A transitions are the only mutation types observed with T>C/A>G being more dominant than C>T/G>A (Appendix A). As homoplasmic mitochondrial mutations are more likely to be maternally inherited mutations or variants arising during early embryonic development, our finding of identical homoplasmic mutations between the two cell types verify that they were derived from the same woman.

### 2.2. SV1 Cells Show Significantly Lower Frequencies of Rare Mutations and Subclonal Mutations than do SV22 Cells

We determined frequencies of rare and subclonal mutations in both cell types by Duplex Sequencing. The overall frequencies of both rare (Figure 1A) and subclonal (Appendix A) mutations are significantly lower in SV1 cells (by 2-fold) than in SV22 cells. In addition, we determined frequencies of each point mutation type, of insertions, and of deletions. C>T/G>A and T>C/A>G transitions are the most prevalent types for both cell types (Figure 1B, Appendix A). Frequencies of each type of rare and subclonal mutations are also significantly lower in SV1 cells than in SV22 cells (Figure 1B, Appendix A).

### 2.3. C>T/G>A Transitions are the Most Prevalent Mutation Types Followed by T>C/A>G in Both Cell Types

The fraction (%) of rare and subclonal mutation types were calculated (Figure 2A, Appendix A). In both SV22 (non-stem) and SV1 (stem) cells, the most prevalent rare and subclonal mutation types are C>T/G>A and T>C/A>G (Figure 2A, Appendix A). The percentages of C>T/G>A and T>C/A>G rare mutations are similar between both cell types. In contrast, the fractions of the four rare mutation types in SV22 and SV1 cells are different by about 1.5-fold with higher fractions C>G/G>C, T>A/A>T, and T>G/A>C mutation types in SV22 cells and higher fractions of C>A/G>T mutation types in SV1 cells (Figure 2A).

To investigate the influences of neighboring bases on types of rare and subclonal mutations, the bases immediately 5′ and 3′ to each mutated base were identified (i.e., the mutation occurs at the second position of each such trinucleotide). This allows classification of observed substitutions into 96 categories (4 bases × 6 substitutions × 4 bases). Numbers and fractions (%) of these mutation trinucleotides in each of the categories compose the “mutation context spectra” (MCS) of the cells. For both rare and subclonal mutations, T>G transversions in context GTA were significantly lower (*p* = 0.02) in SV1 than in SV22 cells (Figure 2B,C, Appendix A). The combined 96 categories of mutation context spectra for both rare (Figure 2B,C) and subclonal (Appendix A) mutations between SV22 and SV1 cells were similar to each other (Cosine similarity scores for rare and subclonal mutations, respectively = 0.98; 0.98).

### 2.4. The Mutation Context Spectra of the Whole mtDNA Rare Mutations in SV22 and SV1 Cells are Compared with the Mutational Signatures of the Catalogue of Somatic Mutations in Cancer (COSMIC)

The mutation context spectra of rare mutations for SV22 and SV1 cells (Figure 2B,C) were compared with each of the COSMIC 30 mutational signatures (https://cancer.sanger.ac.uk/cosmic/signatures) using the Cosine similarity test. The signatures of mutational processes in human cancer, known as “mutational signatures”, are unique combinations of mutation types generated by different mutational processes. To date, a total of 30 mutational signatures are available in COSMIC based on an analysis of 10,952 exomic and 1,048 whole-genomic data across 40 distinct human cancer types. Information such as associated cancer types, proposed etiology, and mutational features are available for each of the 30 signatures.

The top five highly correlated mutational signatures for both SV22 (Figure 3A) and SV1 (Figure 3B) cells are: numbers 5, 30, 16, 19, and 12. The highest correlated signature is No. 5 (Cosine similarity scores of SV22 and SV1 cells: 0.82; 0.79). The signature 5 is found in all cancer types and most cancer samples. The second highest correlated signature is signature 30 (Cosine similarity scores of SV22 and SV1 cells: 0.74; 0.77), which has been observed in breast cancer and osteosarcoma.

### 2.5. The Decreased Mitochondrial Mutagenesis of SV1 Cells Occurs Mainly in the Mitochondrial (mt) RNA Gene Region

The human mitochondrial genome consists of three regions: protein coding-13 genes, RNA (two rRNA subunits and 22 tRNA) genes, and control/overlapped regions. We examined which of the three regions is more vulnerable to mitochondrial mutagenesis in SV22 and SV1 cells.

Of the three categorized regions, the RNA gene regions exhibit the most significant differences in the number of rare (Figure 4A,B) and subclonal (Appendix A) mutations between SV22 and SV1 cells. For example, the number of rare unique mutations in the RNA regions is significantly lower in SV1 cells than in SV22 cells (Figure 4A,B; *p* = 2.6 × 10^−15^). However, the sizes of the three regions are different, contributing to a bias in the number of observed unique mutations. In particular, the number of bases of the protein coding regions (11341) is greater than that of the RNA (4020) and the control/overlapped regions (approximately 1772) [32]. Thus, to compare mutation prevalence per the same genome size between protein-coding regions and RNA regions, we estimated the number of rare unique mutations per 1000 bases and found an even more significantly higher number of rare unique mutations in RNA regions than in the protein coding regions. Within the RNA region itself, there is a significantly lower number of rare unique mutations in SV1 cells than in SV22 cells (Figure 4C,D; *p* = 1 × 10^−4^). This indicates that the RNA regions are more susceptible to mitochondrial mutagenesis than the protein-coding regions.

Genome position and clonality (%) for each of the rare mutations observed in SV22 and SV1 cells are presented in Figure 5A,B. The average and median clonalities (%) of rare mutations are still about 1.7-fold significantly lower in SV1 cells (0.17% ± 0.006) (Figure 5D) than in SV22 cells (0.29% ± 0.007) (Figure 5C), even after excluding variants that were mutated only once (singlet mutations).

### 2.6. Distances between Adjacent Mutations are Longer in SV1 Cells than in SV22 Cells

The average distances between adjacent mutations were calculated for rare mutations present in SV22 and SV1 cells (Figure 6). For each mutation, the number of bases between adjacent mutations, in both 5′ and 3′ directions, were calculated. The minimum and maximum distances, with minimum being distance to the nearest adjacent mutation and maximum being distance to furthest adjacent mutation in either 5′ or 3′ direction, were averaged for each of SV22 and SV1 cells. Both the minimum and maximum average distances to adjacent mutations were observed to be higher in SV1 cells (Min: 8.7 bases; Max: 27.8 bases) than in SV22 cells (Min: 6.9 bases; Max: 21.4 bases) (Figure 6). Statistical significance was observed for both the average minimum and maximum distances to adjacent mutations between SV22 and SV1 cells (*p* < 0.001 by the Mann-Whitney Rank Sum Test). This finding aligns with significantly lower frequencies of rare mutations in SV1 cells than in SV22 cells.

### 2.7. Proportions (%) of Nonsynonymous Mutations in Mitochondrial Protein Coding Regions and Their Predicted Pathogenicity Scores of Subclonal Mutations are Similar between SV22 and SV1 Cells

We compared the proportion (%) of nonsynonymous mutations (causing changes in amino acids) of mitochondrial protein coding regions between SV22 and SV1 cells. Overall, no significant differences in rare or subclonal mutations were observed in the mitochondrial protein coding region between the two cell types (Appendix A).

### 2.8. Predicted Pathogenicity Scores of Missense Mutations of Mitochondrial Protein Coding Regions are Similar between SV22 and SV1 Cells

Predicted pathogenicity scores of the subclonal missense mutations were evaluated using MutPred [33]. MutPred provides a general *g* score for each missense mutation, where a higher *g* score indicates a higher chance that an amino acid substitution is deleterious. The average (and ± SEM) of the *g* score sums for each of the 13 mtDNA protein-coding genes was calculated for subclonal missense mutations present only in SV22 cells (4.25 ± 0.73) and for those present only in SV1 cells (4.15 ± 0.69) but not for those present in both cell types.

### 2.9. Predicted Pathogenicity Scores of Rare Mutations of Mitochondrial tRNA Genes Tend to be Lower in SV1 Cells than in SV22 Cells.

Variants in mt-tRNA genes are a common cause of mitochondrial disease [34]. Thus, we calculated predicted pathogenicity scores for rare and subclonal mutations in mt-tRNA genes using the mitochondrial tRNA informatics predictor (MitoTIP) program of MITOMAP [34]. A higher MitoTIP score represents a higher probability of pathogenicity.

We compared the raw pathogenicity values for rare and subclonal mutations in each of 22 mt-tRNA genes found specifically in SV22 cells vs. in SV1 cells. Eighty-three rare mutations were found in SV22 cells only; while 31 rare mutations were present in SV1 cells only. The exact same mutations specific to each cell type were observed in the subclonal range. The average pathogenicity score of mt-tRNA mutations tends to be lower in SV1 cells than in SV22 cells (Figure 7; *p* = 0.082). In addition, we found four known/confirmed pathogenic tRNA variants (m.5650 G>A, m.5521 G>A, m.5690 A>G, m. 1630 A>G) among the 83 variants in SV22 cells, whereas confirmed pathogenic tRNA variants were not found in SV1 cells.

### 2.10. Duplex Sequencing Identifies Many Rare or Subclonal Mutations that are Specific to SV22 and to SV1 Cells

To compare positions of mutations between SV22 and SV1 cells, only genome positions that had DCS read counts at least 100 or higher in both cell types were considered. Mutations occurring at the same genome positions were scored only once and are referred to as unique mutations. We have identified rare and subclonal unique mutations that are present only in SV22 cells, only in SV1 cells, or in both cell types.

A total of 792 rare and 797 subclonal unique mutations were found in SV22 cells only (but not in SV1 cells); 527 rare and 528 subclonal unique mutations were found in SV1 cells only (but not in SV22 cells) (Appendix A). 447 rare unique mutations were found in both SV22 and SV1 cells, of which 88 are more prevalent in SV22 than in SV1 and 9 are more prevalent in SV1 than in SV22 (prevalence meaning by at least a 3-fold difference) (Appendix A). Similarly, 459 subclonal unique mutations were found in both SV22 and SV1 cells, of which 92 are more prevalent in SV22 than in SV1 and 11 are more prevalent in SV1 than in SV22 by at least a 3-fold difference (Appendix A).

A detailed list of subclonal unique mutations, in which variant reads are at least two or greater, is presented in Appendix A. Mutations were considered singlets if they have a variant read of 1, and these were excluded in Appendix A. A total of 190 non-singlet subclonal mutations were observed only in SV22 cells, but not in SV1 cells, whereas 122 non-singlet subclonal mutations were observed only in SV1 cells, but not in SV22 cells (Appendix A). Among the identified subclonal mutations present in both SV22 and SV1 cells, 92 subclonal mutations were more highly mutated in SV22 cells than in SV1 cells by at least 3-fold (Appendix A). In contrast, only 11 subclonal mutations were more highly mutated in SV1 cells than in SV22 cells by at least 3-fold (Appendix A).

### 2.11. Independent DNA Library Preparation Experiments of Duplex Sequencing (DS) Generate Reproducible Results

Two independent DNA library preparation experiments (Exp1 and Exp2) for DS were performed for both SV22 and SV1 cells. Data from Exp1 and from Exp2 are presented in main text (all Figures and Tables) and in Appendix A, respectively. Consistent results of frequencies (Appendix A) and types (%) (Appendix A) of rare mutations were obtained from the two independent experiments. This validates the reproducibility of our DS data.

## 3. Discussion

Ince et al. (2007) reported that transformation of different normal human breast epithelial cell types led to distinct neoplastic phenotypes. This finding aligns with the observation that the same oncogenes can have quite different phenotypic consequences depending on the cell origin [35,36]. Taken together, these suggest that pre-existing differences in originator normal cell types significantly influence phenotypes of their transformed cells. Gordon et al. (2014) demonstrated that immortalization of human fetal lung fibroblasts increased DNA methylation at gene promoters and caused large-scale changes in gene expression. While these studies [4,37] have examined tumorigenicity, histopathology, and metastatic behavior of human breast transformed cells or DNA methylation and gene expression of immortalized human fibroblasts, no study has examined the subclonal mutations of the whole mitochondrial genome for immortalized derivatives of phenotypically different normal cells.

In the current study, using Duplex Sequencing, we systemically examined rare, subclonal, and clonal mutations of the whole mitochondrial genome for two types of immortalized HBECs [SV22 (non-stem) vs. SV1 (stem)]. Our data indicate that both rare and subclonal mutation frequencies are significantly lower in SV1 cells than in SV22 cells-. Previously, we compared rare mutations of the whole mitochondrial genome between paired normal HBECs (non-stem vs. stem cells) from three independent women and we observed that normal stem cells from two women exhibited significantly lower frequencies of rare mutations than the matching non-stem cells [16]. Our previous [16] and current findings suggest that the significantly low frequencies of rare and subclonal mutations of the whole mitochondria genome in stem cell-derived immortalized (SV1) cells can be attributed to mitochondrial mutations of their parental originator normal cell types. This supports the idea that distinct normal breast epithelial cell types lead to different mitochondrial mutagenesis in their immortalized/transformed cells.

Mechanisms for the lower frequencies of rare and subclonal mutations in stem cell-derived immortalized cells (SV1) than in non-stem cell-derived immortalized cells (SV22) are unclear. A possible mechanism might be associated with lower levels of reactive oxygen species (ROS). The lower ROS levels in stem cells could lead to reduced mtDNA damage and accumulation of rare and subclonal mutations [38,39]. Stem cells might have more efficient systems of DNA repair and mitophagy for removing damaged mtDNA, therefore, lowering the number of mtDNA mutations generated [39,40,41]. These lower ROS levels and reduced mtDNA mutations of normal stem cells could be continually passed down to their immortalized cells.

Our results indicate that rare mutations are more prevalent in the mt-RNA gene regions (mt-RNR1; mt-RNR2; 22 mt-tRNAs) than in the protein coding and control regions. Furthermore, the largest decrease in rare mutation burden in SV1 cells than in SV22 cells is observed in the mt-RNA region. In the mt-RNA regions, the pathogenicity of mt-tRNA mutations can be predicted using the analysis tool MitoTIP. Our results indicate that the lower rare mutation burden of mt-RNA genes in SV1 cells is accompanied by lower predicted pathogenicity of mt-tRNA mutations.

Furthermore, we demonstrate that four known pathological mt-tRNA gene mutations, m.5650 G>A (mt-tRNA^Ala^) [42], m.5521 G>A (mt-tRNA^Trp^) [43], m.5690 A>G (mt-tRNA^Asn^) [44], and m.1630 A>G (mt-tRNA^Val^) [45,46] were present in SV22 cells. However, no known pathogenic mutations were found in the mt-tRNA genes of SV1 cells. Pathogenicities of these four mt-tRNA mutations were confirmed by several studies [42,43,44,45,46]. All four confirmed pathogenic mutations can result in myopathy, a common mitochondrial disease, by impairing protein translation or by causing tRNA instability [42,43,44,45,46]. Cytochrome c oxidase (COX)-deficiency was commonly reported with all four mutations [42,43,44,45,46]. The m.5650 G>A mutation causes a phenotype of pure myopathy [42] and disrupts interaction between mitochondrial alanyl-tRNA synthetase and the mt-tRNAAla aminoacyl acceptor stem. Furthermore, it has been reported to be associated with hepatocellular carcinoma as a result of decreased Complex I and IV activity [47,48]. The m.5521 G>A mutation may impact protein translation [43]. The m.5690 A>G mutation manifests in clinical presentations of chronic progressive external ophthalmoplegia and ptosis [44]. The m.1630 A>G mutation impairs oxygen consumption, which affects the stability of mt-tRNA^Val^ and reduces the levels of subunits of the electron transport chain [46]. Horvath et al. (2009) reported that the m.1630 A>G mutation may cause mitochondrial neurogastrointestinal encephalomyopathy such as gastrointestinal dysmotility and cachexia [45].

In human breast cancer, tRNA mutations can cause significant consequences due to their important roles in protein translation [22]. Mt-tRNA point mutations are typically caused by the loss of mt-tRNA stability, which leads to defective mitochondrial translation and respiratory chain deficiency [44] through: aberrant processing of mRNA transcripts by RNases P and ZL, impaired post-transcriptional mt-tRNA modification such as specific base modifications, 3’-end additions of -CCA sequence and mt-tRNA aminoacylation, as well as compromised interaction of mt-tRNA with both mtEF-Tu (mitochondrial elongation factor Tu) and the mitoribosome [49]. Our results suggest that stem cell-derived immortalized (SV1) cells might possess a more stable mitochondrial genome, exhibited by lower subclonal mutation burden, longer distances to adjacent mutations, and lower predicted pathogenicity than non-stem cell-derived immortalized (SV22) cells.

In summary, we examined mutational profiles of immortalized cells derived from normal human breast epithelial cells of different phenotypes (non-stem vs. stem) and have associated them with mutational profiles of phenotypically distinct normal HBECs, suggesting influences of originator normal cell phenotypes on determining mutational profiles of their immortalized derivatives. Our results indicate that the vast majority of mutations in the cells are rarely occurring mutations, which are not detectable by conventional DNA sequencing methods, but are accurately detectable by Duplex Sequencing. The most prevalent rare mutation types are C>T/G>A and A>G/T>C transitions. Immortalized stem cells (SV1) exhibit lower frequencies of rare mutations than immortalized non-stem cells (SV22), which is supported by the higher average distances between adjacent rare mutations in SV1 than in SV22. The reduced mitochondrial rare mutation burden of immortalized stem cells mainly occurs in RNA gene regions of the whole mtDNA, and these are accompanied by reduced predicted pathogenicity. Our findings suggest that phenotypes of originator parental normal cells significantly influence and direct mutational profiles of the whole mitochondrial genome in their immortalized derivatives. Our results have implications in investigating genetic changes of mitochondrial genomes acquired during cellular immortalization and in characterizing immortalized stem (vs. non-stem) cells, which represent in vitro preneoplastic stages of breast carcinogenesis.

## 4. Materials and Methods

### 4.1. Development and Culture of Immortalized Human Breast Epithelial Cells

Both Type-I (stem cell features) and Type-II (without stem cell features) normal HBECs were transformed with SV40-T into immortalized cells, M13SV1 and M13SV22, respectively. These immortalized cells were provided by Dr. Chia-Cheng Chang at Michigan State University (MSU) in East Lansing, MI, USA. A Material Transfer Agreement was approved by both MSU and University of Washington (UW). Development, characterization, and culture of normal HBEC and *in vitro* transformed HBECs were described previously [7,9,10,11,12,16,50,51,52,53,54].

### 4.2. DNA Extraction, Adapter Synthesis, and DNA Library Preparation for Duplex Sequencing (DS)

DNA extraction, adapter synthesis, and DNA library preparation for DS were carried out as previously described [16].

### 4.3. Data Processing

DS data were processed as described previously [16,23,24,55] with modifications. Previously, our in-house DS script processed and aligned two read files of pair-end sequencing data separately, then merged the two data streams. For the current study, the script was modified to merge the two sequence read files first then process and align the merged file. This modification improved accuracy and efficiency of data processing. Sequence reads were aligned to the revised Cambridge Reference sequence (rCRS) reference genome (NC_012920) using BWA and the genome analysis toolkit (GATK) software as described previously [16]. BWA “mem” was used in replacement of BWA “aln”. During processing, all dataset reads were filtered using a mapping quality score of 40. Pileup-based variant calling used the default base quality score of 13. The first four bases at 5′ and 3′ ends of each sequence reads were clipped to eliminate potential artifactual variants commonly present at the ends of each read.

### 4.4. Data Analysis for Positions, Frequencies, Types, and Context Spectra of Mutations and Annotation of Genetic Variants

Genome positions with a DCS sequence read depth of 100 or greater were included for calculating mutation frequencies. For all other data analysis, only genome positions that had a minimum DCS read of 100 in matching positions of both samples (SV22; SV1) were considered. The total number of DCS variant reads were divided by total number of DCS sequenced reads to calculate mutation frequencies for each sample. The total number of unique mutations (i.e., mutants were scored only once at each position of the genome regardless of number of variant reads observed in that position) were used for all other analyses, which include fractions (%) of mutation types, mutation context spectra (%), and comparison of mutation positions. Mutations of different types that occur at a same position (i.e., a wildtype nucleotide base at a specific genome position is mutated to two different variant bases) were considered as two different unique mutations. In addition, mutations in positions transcribed into two different protein-coding genes or RNA genes (i.e., regions where two protein coding genes or RNA gene overlap) were also considered as two different unique mutations. Genetic variant data from DS were annotated using Annotate Variation (ANNOVAR) bioinformatical software version 2017Jun01 (annovar.openbioinformatics.org) [56]. Annotations were added for each sequenced position of the whole mitochondrial genome.

### 4.5. Cosine Similarity Analysis and Comparison of Mutation Context Spectra with the COSMIC Mutational Signatures

Cosine similarity (https://neo4j.com/docs/graph-algorithms/current/algorithms/similarity-cosine) can be used to compare mutation context spectra [57]. Cosine similarities between the mutation context spectra of SV22 (immortalized non-stem) and SV1 (immortalized stem) cells were computed by considering each spectra a 96 tuple ‘vector’, and then dividing the inner product of each of the two vectors by the product of their magnitudes, giving a ‘cosine’ in a 96-dimensional space. Since each spectrum consists of non-negative values only (vectors lie in the ‘first quadrant’) these cosines range from 1 (identical pattern) to 0 (no common contexts). This technique considers every mutation-context completely independent of any other context. For example, G[C>T]T has zero similarity with both G[C>T]A and A[T>G]C. (Note: Soft cosine similarity allows for a degree of ‘sameness’ between various contexts).

In addition, cosine similarities were further computed between the rare mutation context spectra of SV22 and SV1 cells and the 30 COSMIC mutational signatures identified so far (https://cancer.sanger.ac.uk/cosmic/signatures).

### 4.6. Analysis of Distances to Adjacent Mutations

The distances between adjacent mutations, in bases, were measured for each rare mutation in SV22 and SV1 cells. For each rare mutation, a distance was measured to adjacent mutations in both 5′ and 3′ directions. The distance to the closer adjacent mutation was considered as minimum distance and that to the further adjacent mutation was consider the maximum distance. Minimum and maximum distances of rare mutations in the whole mtDNA genome were each averaged for both SV22 and SV1 cells. For calculating distances, all mutations at each specific genome position were considered to be one unique mutation regardless of mutated nucleotide type, since the mutation type does not influence distance.

### 4.7. Predicted Pathogenicity

The MutPred program version 2.0 (http://mutpred.mutdb.org) [33] was used to obtain pathogenicity scores of missense mutations in mt-protein coding regions as described previously [28,48]. The predicted pathogenicity scores for mt-tRNA mutations were analyzed using MitoTIP (August 2017 version) available via MITOMAP (www.mitomap.org). MitoTIP analysis [34] involves calculating pathogenicity scores for each possible variant through a summation of variant history, conservation score, position score, and secondary structure score. Using publicly-available databases, an algorithm that estimates the importance of a position across mtDNA was generated and used in scoring. Pathogenicity scores ranged from −5.9 to 21.8 and were assigned to each variant in mt-tRNAs by its position.

### 4.8. Statistical Analysis

Differences in mtDNA mutation frequencies and in the fraction (%) of mutation types between the two groups were analyzed by performing the prop.test for 2-sample equality of proportions with continuity correction (also called Chi-Square test) using an R program (version 3.4.4). To compare the MitoTIP predicted pathogenicity scores between the two groups, the Mann-Whitney U-test (Wilcoxon Rank-Sum test) was applied using Sigma Plot (version 12.0) (Systat Software, San Jose, CA, USA). Differences between the two groups were considered significant if the *p*-value was less than 0.05.

## Figures and Tables

**Figure 1 ijms-20-02813-f001:**
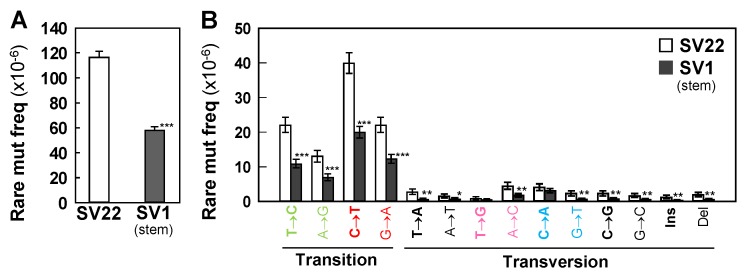
Frequencies of rare mutations in the whole mtDNA. Overall rare mutation frequency (**A**) and frequencies of rare mutation types (**B**) for SV22 (immortalized non-stem) and SV1 (immortalized stem) cells were determined using Duplex Sequencing. Error bars represent the Wilson Score 95% confidence intervals. Significant differences in rare mutation frequencies between two groups are indicated (* *p* < 0.05, ** *p* < 5 × 10^−4^, and *** *p* < 5 × 10^−10^) by the Chi-Square test.

**Figure 2 ijms-20-02813-f002:**
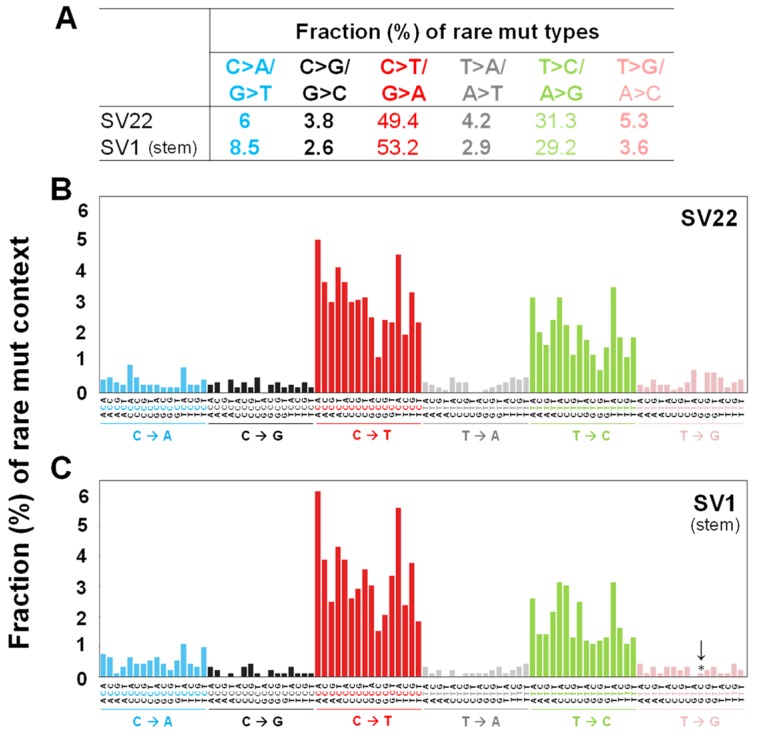
Types and sequence context spectra of rare unique mutations in the whole mtDNA. Fractions (%) of rare mutation types (**A**) and fractions (%) of rare mutation context spectra (**B**,**C**) for SV22 (immortalized non-stem) and SV1 (immortalized stem) cells were determined using Duplex Sequencing. Trinucleotide contexts (**B**,**C**) are the mutated base surrounded by all possibilities for its immediate 5′ and 3′ bases. To keep the graph concise, these point mutation trinucleotides are complemented as necessary to always depict the reference base as the pyrimidine of its pair. The fraction (%) of each specific trinucleotide out of all 96 possible trinucleotide contexts depicts the contribution of each genome sequence context to each point mutation type. Significant differences in fractions (%) of mutation context types between the two groups are indicated (* *p* < 0.05) by the Chi-Square test.

**Figure 3 ijms-20-02813-f003:**
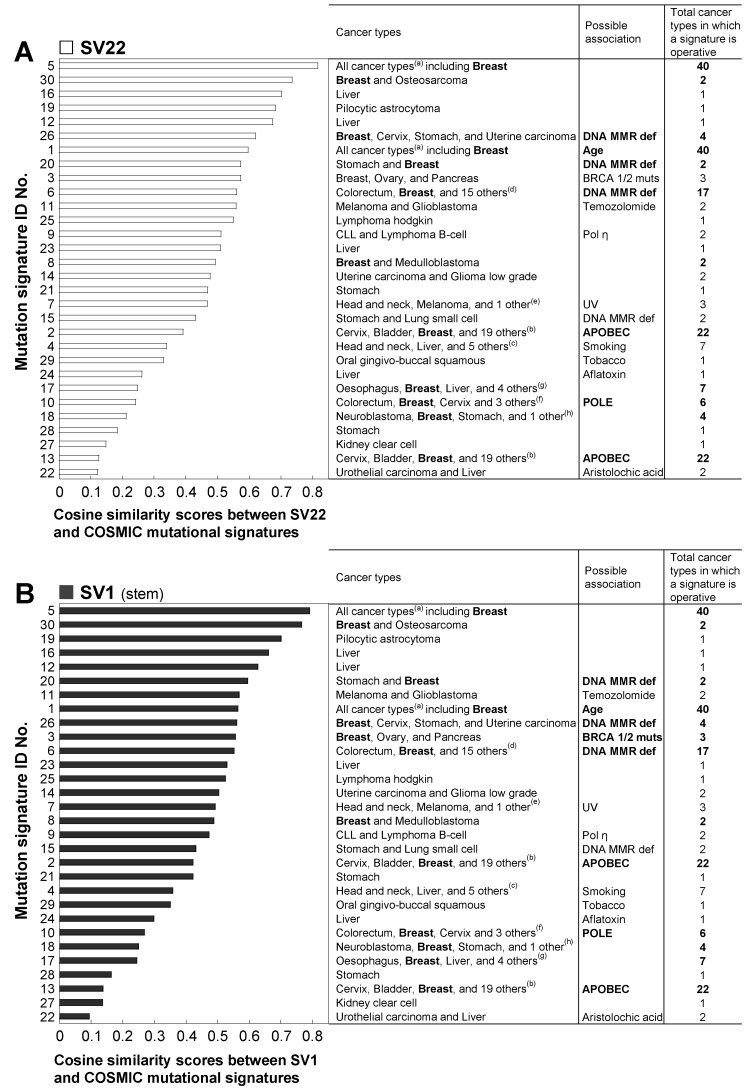
Correlation between the mutation context spectra of the mtDNA rare unique mutations in SV22 (immortalized non-stem) (**A**) and SV1 (immortalized stem) (**B**) cells vs. the 30 published COSMIC mutational signatures across the spectrum of human cancer types. The cosine similarity test was used to correlate SV22 and SV1 cells against the 30 published COSMIC mutational signatures (https://cancer.sanger.ac.uk/cosmic/signatures). ^(a)^ Adrenocortical carcinoma, ALL, AML, Bladder, Breast, Cervix, Chondrosarcoma, CLL, Colorectum, Glioblastoma, Glioma low grade, Head and neck, Kidney chromophobe, Kidney clear cell, Kidney papillary, Liver, Lung adenocarcinoma, Lung small cell, Lung squamous, Lymphoma B-cell, Lymphoma hodgkin, Medulloblastoma, Melanoma, Myeloma, Nasopharyngeal carcinoma, Neuroblastoma, Oesophagus, Oral gingivo-buccal squamous, Osteosarcoma, Ovary, Pancreas, Paraganglioma, Pilocytic astrocytoma, Prostate, Stomach, Tyroid, Urothelial carcinoma, Uterine carcinoma, Uterine carcinosarcoma, Uveal melanoma; ^(b)^ Adrenocortical carcinoma, ALL, CLL, Head and neck, Kidney papillary, Lung adenocarcinoma, Lung squamous, Lymphoma B-cell, Myeloma, Nasopharyngeal carcinoma, Oesophagus, Oral gingivo-buccal squamous, Osteosarcoma, Pancreas, Stomach, Tyroid, Urothelial carcinoma, Uterine carcinoma, Uterine carcinosarcoma; ^(c)^ Adenocortical carcinoma, Lung adenocarcinoma, Lung squamous carcinoma, Lung small cell, and Oesophagus; ^(d)^ Adrenocortical carcinoma, Cervix, Glioma low grade, Kidney chromophobe, Kidney clear cell, Liver, Lung adenocarcinoma, Nasopharyngeal carcinoma, Oesophagus, Osteosarcoma, Pancreas, Prostate, Uterine carcinoma, Uterine carcinosarcoma, Uveal melanoma; ^(e)^ Oral gingiva-buccal squamous; ^(f)^ Bladder, Uterine carcinoma, and Uterine carcinosarcoma; ^(g)^ Lung adenocarcinoma, Lymphoma B-cell, Stomach, and Melanoma; ^(h)^ Adrenocortical carcinoma. Abbreviations used are: ALL, acute lymphoblastic leukemia; AML, acute myeloid leukemia; CLL, chronic lymphocytic leukemia; Def, deficiency; MMR, mismatch repair; Muts, mutations; Pol, polymerase; UV, ultraviolet light.

**Figure 4 ijms-20-02813-f004:**
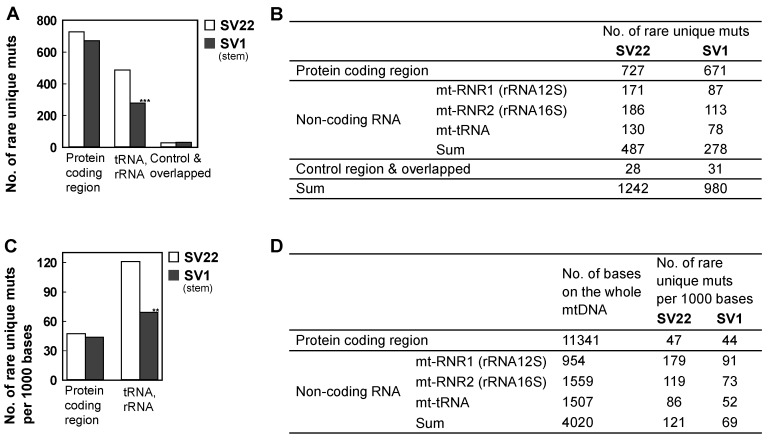
Number of rare unique mutations considered (individually and per 1000 bases) by genome positional category in the whole mtDNA. Number of rare unique mutations by positional category (**A**,**B**) and number of rare unique mutations per 1000 bases by positional category (**C**,**D**) were calculated for SV22 (immortalized non-stem) and SV1 (immortalized stem) cells. Only genome positions that had a DCS read depth of ≥100 in both samples were considered. Significant differences in numbers of rare mutations between the two groups are indicated (* *p* < 0.05, ** *p* < 5 × 10^−4^, and *** *p* < 5 × 10^−10^) by the Chi-Square test.

**Figure 5 ijms-20-02813-f005:**
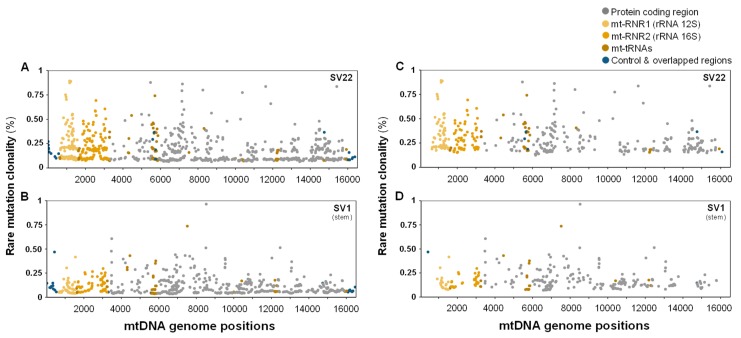
Genomic positions and clonalities (%) of rare mutations in the whole mtDNA. Rare mutation clonalities (%) by genomic position including singlets (**A**,**B**) or excluding singlets (**C**,**D**) were determined for SV22 (immortalized non-stem) and SV1 (immortalized stem) cells using Duplex Sequencing. Singlets are defined as variants that are mutated only once in nucleotides sequenced at a specific genome position.

**Figure 6 ijms-20-02813-f006:**
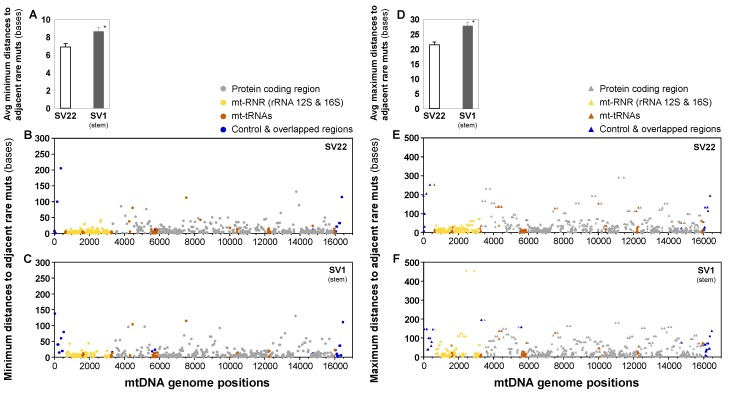
Genomic positions and average minimum and maximum distances to adjacent rare mutations (in bases) in the whole mtDNA. Minimum (**A**–**C**) and maximum (**D**–**F**) distances to adjacent rare mutations in both 5′ and 3′ directions were determined for each SV22 (immortalized non-stem) and SV1 (immortalized stem) rare mutations using Duplex Sequencing. Error bars represent standard error of the mean (SEM). Significant differences in average distances to adjacent rare mutations between two groups are indicated (* *p* < 0.001) by the Mann-Whitney Rank Sum Test.

**Figure 7 ijms-20-02813-f007:**
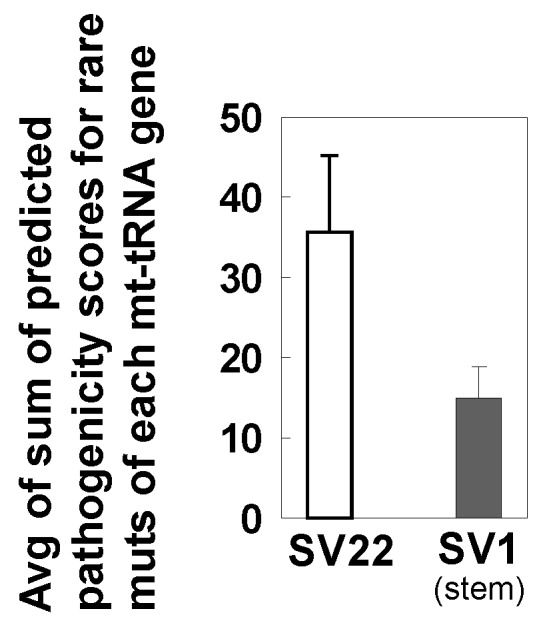
Predicted pathogenicity of rare unique mutations of mt-tRNAs. Predicted pathogenicity scores of rare mutations in mt-tRNAs were obtained using MitoTIP and then were totaled for each region of tRNAs. Only the mutations present exclusively in each sample (mt-tRNA mutations present only in SV22 cells vs. mt-tRNA mutations present only in SV1 cells) were included. The sums of predicted pathogenicity scores from each mt-tRNA region were averaged. Error bars represent the standard error of the mean (SEM).

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
