# Peer review of "Immortalization of Different Breast Epithelial Cell Types Results in Distinct Mitochondrial Mutagenesis"

_ijms, 2019, doi:10.3390/ijms20112813_

Round 1
Reviewer 1 Report
This is an interesting and well-written study that looks at the impact of lineage (stem-cell vs non-stem cell) on mitochondrial mutagenesis following transformation in vitro. It is novel and the data support the main conclusions, although the overall scope is mostly descriptive. The manuscript would greatly benefit from addressing the following points aiming at extracting the most mechanistic information possible:
1. The rationale needs to be explained in the introduction. why were mitochondrial genomes selected? what is the functional significance of finding deleterious mutations in non-coding vs. protein-coding DNA?
2. The mechanistic information that can be derived from the data has not been fully exploited: what known mutation signature (COSMIC signatures, Alexandrov et al Nature 2013) does the profile shown in Fig. 2 resemble most? Even if it doesn't match any known signature, the authors need to state that they did the analysis. What can be said about the fact that the profile is strand-specific? And about the average distance/clustering of the mutations? This is all mechanistic information that is not exploited. Are the differences consistent with differences in the level of ROS damage?
3. Overall, the differences seen between the two cell types (SV1 and SV2) are small (in the order of 1.5-2 fold) both in terms of frequency and in terms of pathogenicity. The identification of key pathogenic mutations in SV2 that are absent in SV1 appears to be more significant and should be mentioned in the abstract. Was this also the case in the previous study comparing rare 267 mutations of the whole mitochondrial genome between paired normal HBECs (non-stem vs stem 268 cells) from three independent women? Is it consistent with the observed difference in mutatio spectrum between SV1 and SV2.
4. The statistics to determine differences between mutation spectra may be inadequate. Cosine similarity (a modified Manhattan distance) should be used to compute the differences between two spectra.
Author Response
[1-A]. The rationale needs to be explained in the introduction. why were mitochondrial genomes selected?
what is the functional significance of finding deleterious mutations in non-coding vs. protein-coding DNA?
We thank Reviewer-1’s positive comments and suggestion. The introduction has been updated as suggested
by the reviewer (Revision p.2: Lines 69-79)
[1-B]. The mechanistic information that can be derived from the data has not been fully exploited: what known
mutation signature (COSMIC signatures, Alexandrov et al Nature 2013) does the profile shown in Fig
2 resemble most? Even if it doesn't match any known signature, the authors need to state that they did the
analysis. What can be said about the fact that the profile is strand-specific? And about the average
distance/clustering of the mutations? This is all mechanistic information that is not exploited. Are the
differences consistent with differences in the level of ROS damage?
We conducted additional analyses to compare the mutation context spectra of SV22 and of SV1 cells (Fig 2)
with the latest COSMIC signatures (https://cancer.sanger.ac.uk/cosmic/signatures) and the new data have
been added in Results (Revision Fig 3; p.4-6: Lines 158-198). Methods have been updated accordingly
(Revision p.12: Lines 443-458). The strand-specificity was not the main finding or focus of our study, so
references to it (Initial version: Fig 1C and 1D) have been removed. Instead, we have performed new analyses
(Revision Fig 3, Fig 6) suggested by Reviewer-1 to better focus and improve our manuscript. The analysis
on the average distances between mutations has been done and is included in Results (Revision Fig 6; p.7-
2
8: Lines 234-252) and Methods have been updated accordingly (Revision p.12-13: Lines 459-467).
Changes of ROS levels have been discussed as a possible mechanism for lower mutation burdens in SV1
cells than in SV22 cells (Original manuscript p.8: Lines 275-282; Revision p.10: Lines 343-350).
Please see the list of rearranged figures and tables below.
Original manuscript Revision Fig #, Table #
Fig 1A, 1B Same
Fig 1C, 1D Delete
Fig 2A, 2B, 2C Same
New Fig Fig 3A, 3B (New data: Cosine similarity)
Fig 3A Fig 4A
Table S1A Fig 4B
Fig 3B Fig 4C
Table S2 Fig 4D
Fig 4A - 4D Fig 5A - 5D
New Fig Fig 6A -6F (New data: Mutation distance)
Fig 5 Fig 7
Fig S1A - S1D Same
Fig S2A, S2B Same
Fig S2C, S2D Delete
Fig S3A, S3B, S3C Same
Fig S4A, S4B (2 bars each) Fig S4A, S4B (New data:SV22, 4 bars each)
Fig S4C, S4D Delete
Table S1B Table S1
Table S3 Table S2
Table S4 Table S3
Table S5 Table S4
[1-C]. Overall, the differences seen between the two cell types (SV1 and SV22) are small (in the order of 1.5
to 2 fold) both in terms of frequency and in terms of pathogenicity. The identification of key pathogenic
mutations in SV22 that are absent in SV1 appears to be more significant and should be mentioned in the
abstract. Was this also the case in the previous study comparing rare mutations of the whole mitochondrial
genome between paired normal HBECs (non-stem vs stem cells) from three independent women? Is it
consistent with the observed difference in mutation spectrum between SV1 and SV22?.
We have updated the abstract by including some of the known key pathogenic mutations identified in SV22
but not in SV1 (Revision p.1: Lines 28-30). Compared to our previous study on normal HBECs (stem vs nonstem),
the current study on immortalized cells show greater and more statistically significant differences in
rare mutation frequency between the two different cell types. The observed differences in the rare mutation
frequency for immortalized stem vs immortalized non-stem cells show that phenotypes of originator normal
cells may have an influence on the mitochondrial mutation profiles of transformed cells.
[1-D]. The statistics to determine differences between mutation spectra may be inadequate. Cosine similarity
(a modified Manhattan distance) should be used to compute the differences between two spectra.
We have computed Cosine similarity analyses between SV1 and SV22 mutation context spectra and the 30
COSMIC signatures. Our Results (Revision p.4: Lines 143-146; Fig 3; p.4-6: Lines 158-198) and Methods
(Revision p.12: Lines 443-458) have been updated accordingly. Please see the list of rearranged figures
and tables above [1-B].
Reviewer 2 Report
This manuscript describes an interesting finding whereby transformed breast cells, stem versus non-stem, of different tumorigenic phenotypes have diverse mitochondrial DNA rare mutations. Importantly, it was observed that stem cell derived transformed cells (SV1) have significantly less rare mtDNA mutations than non-stem cell derived transformed cells (SV22). The majority of these rare mtDNA mutations are found within non-coding DNA regions and correlate with the pathogenicity of these cells.
Although the findings in this study are interesting, the study appears mostly observational and does not address the biological significance of these findings in terms of mitochondrial function or how these mtDNA mutations may contribute to the diverse phenotype of SV1 compared to SV22 cells. Given the significant increase in mtDNA mutations in SV22 cells compared to SV1, it would have been relevant to design experiments that address whether these mutations impose any biological phenotype, for example do these mutations have any effect on mitochondrial function? and does this contribute to the difference in tumorigenic phenotype?
It was not clear as to why the focus in the main text was on rare mutations. All data for sub clonal mutations were in supplemental.
There are no error bars in the graphs of Figure 3, thus it's not clear how statical analysis was performed.
Relevant tables should be included in main body of manuscript.
In section 2.10, it is stated that "Two independent DNA library preparation experiments were performed for SV1 cells. Consistent results of frequencies (Fig S4A), types (%) (Fig S4B), and sequence context spectra (%) of rare mutations (Fig S4C-D) were obtained from the two independent experiments. This validates the reproducibility of our Duplex Sequencing data." Why was this not also done for SV22 cells. It is equally important to have replicate analysis for SV22 cells.
From my understanding, the purpose of this study was to adress whether there are changes in mutations, in the mitochondrial genome level, between transformed and originator cells of different phenotypes. The authors conclude that immortalization of these cells does not contribute to the mutations, that the mutations are passed on from the originator cells. To truly make this claim it would be important to perform sequencing analysis on the originator cells, within the same study, followed by a comparative analysis of the mutations in the originator and transformed cells. This would also allow us to see if the same mutations are maintained or if new ones are formed during the transformation process. As it is now, a possible interpretation of the results is that the non-stem cells are more prone to mutations during immoratlization.
In the discussion the authors refer us to a previous study where they looked at the mitochondrial genome in non-immortalized HBEC non-stem vs stem cells, and made a similar observation to this study, whereby the non-stem cells had more mtDNA mutations than the stem cells. Thus it is not clear what new information was gained from this study. Again here, a comparative analysis would have been ideal, both as a control for this study, as well as a means to identify whether the same mtDNA mutations in non-transformed cells are maintained following transformation.
Again in the discussion, authors make the following statement: "In summary, we examined whether different phenotypes of originator normal human breast epithelial cells (non-stem vs stem) can lead to different profiles of mitochondrial mutations for their immortalized derivatives." This is not an accurate statement given that in this study there was no comparison made between originator and immortalized cells.
It is not clear why authors discuss the relevance of the identified mtDNA mutations in myopathies in the discussion section. It would be more relevant to draw associations in a cancer setting.
Author Response
[2-A]. Although the findings in this study are interesting, the study appears mostly observational and does not address the biological significance of these findings in terms of mitochondrial function or how these mtDNA mutations may contribute to the diverse phenotype of SV1 compared to SV22 cells. Given the significant increase in mtDNA mutations in SV22 cells compared to SV1, it would have been relevant to design experiments that address whether these mutations impose any biological phenotype, for example do these mutations have any effect on mitochondrial function? and does this contribute to the difference in tumorigenic phenotype?
We thank Reviewer-2 for the comments. The focus of this study was to examine influences on rare and subclonal mtDNA mutational profiles of immortalized cells as a result of differences in phenotypes of their originator normal cells. We have previously examined differences in rare and subclonal mtDNA mutational profiles between normal stem HBECs and normal non-stem HBECs. In comparison, we designed the current study to examine if the differences in phenotypes of originator normal cells have an influence on mutational profiles of the whole mitochondrial genome in transformed/immortalized cells. We conducted comprehensive analyses for mutations of the whole mtDNA in immortalized cells derived from phenotypically different parental normal cells. Importantly, we have applied Duplex Sequencing to detect not only high-frequency mutations but also low-frequency mutations with unprecedented accuracy. Considering the scope and multiple comprehensive mutation analyses that the current study accomplished, the high cost of Duplex Sequencing, and the time required for the current study, we feel studies on the functional effects of the identified rare mutations must be undertaken in future experiments.
Furthermore, please note that we have updated and strengthened this manuscript by adding new analysis results: Distances of mutations (Revision Fig 6; p.7-8: Lines 234-252); Cosine similarity for comparing our mutation context spectrum results with the latest COSMIC signatures (Revision Fig 3; p.4-6: Lines 158-198); and Independent Duplex Sequencing experimental results of SV22 cells (Revision Fig S4; p.10: Lines 311-317).
4
[2-B]. It was not clear as to why the focus in the main text was on rare mutations. All data for sub clonal mutations were in supplemental.
The main text was focused on rare mutations because most of the identified subclonal mutations occurred at rare mutation allele frequencies (%). In this study, we defined mutations as rare (0-1%) or subclonal (0-10%) based on their allele frequency. Of the 1256 subclonal mutations identified in SV22, 1242 mutations had allele frequency less than or equal to 1%. Similarly, of the 988 subclonal mutations identified in SV1, 980 mutations had allele frequency less than or equal to 1%. Thus, because almost all subclonal mutations occur at rare allele frequencies and rare mutations are not well studied due to high error rates of conventional DNA sequencing methods, we have focused on rare mutations and presented rare mutation results in the main text of the manuscript. Subclonal mutation results are presented as supplementary materials.
[2-C]. There are no error bars in the graphs of Figure 3, thus it’s not clear how statistical analysis was performed.
The data presented by Figure 3 (Revision Fig 4; p.6-7: Lines 199-215; 220-227) are counts of rare unique mutations identified in each of mtDNA regions (protein coding, tRNA & rRNA, and Control & overlapped regions). Therefore, no error bars exist. To clarify Fig 3A and 3B, we have incorporated Table S1A and Table S2 into Fig 3 (Revision Fig 4), respectively. Please see below for the revised Figure and Table numbers for these changes.
Original manuscript
Revision Fig #, Table #
Fig 1A, 1B
Same
Fig 1C, 1D
Delete
Fig 2A, 2B, 2C
Same
New Fig
Fig 3A, 3B (New data: Cosine similarity)
Fig 3A
Fig 4A
Table S1A
Fig 4B
Fig 3B
Fig 4C
Table S2
Fig 4D
Fig 4A - 4D
Fig 5A - 5D
New Fig
Fig 6A -6F (New data: Mutation distance)
Fig 5
Fig 7
Fig S1A - S1D
Same
Fig S2A, S2B
Same
Fig S2C, S2D
Delete
Fig S3A, S3B, S3C
Same
Fig S4A, S4B (2 bars each)
Fig S4A, S4B (New data:SV22, 4 bars each)
Fig S4C, S4D
Delete
Table S1B
Table S1
Table S3
Table S2
Table S4
Table S3
Table S5
Table S4
The statistical analysis for Fig 3A and 3B (Revision Fig 4A and 4C; p.6-7: Lines 199-215; 220-227) was performed by conducting a Chi-square test (2-sample test for equality of proportions with continuity correction) using the identified number of rare unique mutations for each region and the total number of rare unique mutations in the whole mtDNA. For example, in Fig 3A (Revision Fig 4A), we compared SV22 cells’ 186 mt-tRNA rare mutations (out of 1242 in the whole mtDNA) to SV1 cells’ 113 mt-tRNA rare mutations (out of 980 in the whole mtDNA). In Fig 3B (Revision Fig 4C), 86 mt-tRNA rare mutations per 1000 bases (out of 121
5
per 1000 bases in the whole mtDNA) in SV22 were compared with 52 mt-tRNA rare mutations per 1000 bases (out of 69 in the whole mtDNA) in SV1. The use of the Chi-square test was based on a consultation with a statistician.
[2-D]. Relevant tables should be included in main body of manuscript.
Relevant tables are now included in main body of manuscript. Please see the list of rearranged figures and tables above [2-C].
[2-E]. In section 2.10, it is stated that "Two independent DNA library preparation experiments were performed for SV1 cells. Consistent results of frequencies (Fig S4A), types (%) (Fig S4B), and sequence context spectra (%) of rare mutations (Fig S4C-D) were obtained from the two independent experiments. This validates the reproducibility of our Duplex Sequencing data." Why was this not also done for SV22 cells. It is equally important to have replicate analysis for SV22 cells.
We did independent DNA library preparation experiments for both SV1 and SV22 cells. As independent experimental data of SV1 clearly show similar results, we initially did not include independent experimental data of SV22 cells in the original manuscript. However, in the revised manuscript, we have added the independent results of SV22 cells as suggested by the reviewer (Revision Fig S4; p.10: Lines 311-317). In addition, the reproducibility and accuracy of Duplex Sequencing have been demonstrated in previous studies by us [Kennedy et al 2014, Ahn et al 2015, Ahn et al 2016, Ahn and Lee 2019] and others [Schmitt et al 2012, Kennedy et al 2013, Schmitt et al 2015, Krimmel et al 2016, Reid-Bayliss et al 2016].
[2-F]. From my understanding, the purpose of this study was to address whether there are changes in mutations, in the mitochondrial genome level, between transformed and originator cells of different phenotypes. The authors conclude that immortalization of these cells does not contribute to the mutations, that the mutations are passed on from the originator cells. To truly make this claim it would be important to perform sequencing analysis on the originator cells, within the same study, followed by a comparative analysis of the mutations in the originator and transformed cells. This would also allow us to see if the same mutations are maintained or if new ones are formed during the transformation process. As it is now, a possible interpretation of the results is that the non-stem cells are more prone to mutations during immortalization.
We concluded our study by stating that “phenotypes of originator parental normal cells significantly influence and direct mutational profiles of the whole mitochondrial genome in their immortalized derivatives”. We do not conclude that immortalization of the cells does not contribute to the mutations.
Although the originator normal cells from the same woman were not directly compared with the matching immortalized cells within the present study, the results of our current study clearly indicate that differences in mitochondrial mutation profiles exist between the two immortalized cell types of different phenotypes. The two immortalized cell types were cultured and prepared under identical conditions and the only difference between the two are their originating normal cell phenotypes, suggesting that the phenotype of originating normal cells influences the mitochondrial mutation profiles of their immortalized derivatives.
It is possible that non-stem cells are more prone to mitochondrial mutations during immortalization as the reviewer suggested. We are open to such a possibility and do not rule it out. However, if the mitochondrial genome of non-stem cells are more prone to mutations, thereby increasing rare and subclonal mutation frequency of SV22 compared to that of SV1, then one would conclude that the immortalized/transformed cells’
6
mutational profiles have been influenced by differences in phenotypes of their originator normal cells. This is a restatement of our conclusion.
Regardless, we believe that it is more likely that the differences in mitochondrial mutational profiles of immortalized cells are due to differences in their originating parental normal cells. In addition, our previous study demonstrated lower mtDNA rare mutation frequencies in normal stem cells compared to the frequencies in normal non-stem cells of the matching women [Ahn et al. 2015]. The cells used in the previous study [Ahn et al. 2015] and the ones used in the current study were cultured and collected under similar conditions by the same research personnel using same experimental protocols. We predict that similar differences in rare and subclonal mtDNA mutation frequencies would be observed in the parental normal cells of SV22 and SV1 cells had we done analyses with them, and that differences in rare and subclonal mtDNA mutational profiles of phenotypically distinct normal cells lead to similar differences of mutational profiles after transformation.
Lastly, considering the scope and comprehensive mutation analyses that the current study accomplished, the high cost of Duplex Sequencing, and the time/resources required for conducting the comprehensive deep sequencing analyses and supporting results from our previous and current studies, we feel studies on the matching normal cells must be undertaken in separate future experiments. In addition, please note that we have updated and strengthened this manuscript by adding new analysis results: Distance of mutations (Revision Fig 6; p.7-8: Lines 234-252); Cosine similarity for comparing our mutation context spectrum results with the latest COSMIC signatures (Revision Fig 3; p.4-6: Lines 158-198), and independent Duplex Sequencing experimental results of SV22 cells (Revision Fig S4; p.10: Lines 311-317).
[2-G]. In the discussion the authors refer us to a previous study where they looked at the mitochondrial genome in non-immortalized HBEC non-stem vs stem cells, and made a similar observation to this study, whereby the non-stem cells had more mtDNA mutations than the stem cells. Thus, it is not clear what new information was gained from this study. Again here, a comparative analysis would have been ideal, both as a control for this study, as well as a means to identify whether the same mtDNA mutations in non-transformed cells are maintained following transformation.
Previous studies, such as Ince et al (2007), demonstrated that the transformation of different originator normal cell phenotypes leads to distinct tumor phenotypes. However, no study has examined differences in rare and subclonal mutational profiles of the whole mitochondrial genome between transformed/immortalized human cells derived from phenotypically different normal cells. We aimed to identify differences in rare and subclonal mutations between two immortalized cell types derived from phenotypically distinct originator normal cells. We characterized rare and subclonal mtDNA mutational profiles of immortalized SV22 and SV1 cells by investigating the frequencies, types, context spectra, positions, distances, and predicted pathogencity of their mutations. We also identified specific mutations that are exclusively found in SV22 vs SV1 cells and mutations found in both SV22 and SV1 cells (Original manuscript Table S5, p.7: Lines 238-245); (Revision Table S4; p.9: Lines 303-310).
Please note that we have conducted many new important analyses in the present study, which were not investigated in our previous studies. The new analysis we performed and bioinformatic techniques we employed in the present study are to: Determine in which regions of the whole mtDNA are more susceptible to mutations (Revision Fig 4; p.6-7: Lines 199-215; 220-227); Examine the predicted pathogenicity of mt-tRNA mutations (Revision Fig 7; p.8-9: Lines 267-289); Compute distances between adjacent mutations (Revision Fig 6; p.7-8: Lines 234-252); Compare our mutation context results between the two cell types using Cosine Similarity analysis (Revision pg.4: Lines 144-146) and correlate them with COSMIC signatures (Revision Fig 3; p.4-6: Lines 158-198).
7
[2-H]. Again, in the discussion, authors make the following statement: "In summary, we examined whether different phenotypes of originator normal human breast epithelial cells (non-stem vs stem) can lead to different profiles of mitochondrial mutations for their immortalized derivatives." This is not an accurate statement given that in this study there was no comparison made between originator and immortalized cells.
As we have explained above in [2-F], we do believe that the phenotypic difference in the originator normal cells leads to different mitochondrial mutation profiles of their transformed cells. SV22 and SV1 cells used in this study were prepared for Duplex Sequencing under the same conditions, and only difference between the two cell types were the phenotypes of their originator normal cells. Although a direct comparison was not made between the immortalized cells and their originator normal cells, the differences in rare and subclonal mtDNA mutation profiles between phenotypically distinct normal HBECs have been observed in previous studies. Therefore, an association can now be made between differences in mtDNA mutational profiles of phenotypically distinct cells before and after transformation. However, in order to more accurately represent our findings and conclusion, we have rewritten the concluding sentence (Revision p.11: Lines 385-389).
[2-I]. It is not clear why authors discuss the relevance of the identified mtDNA mutations in myopathies in the discussion section. It would be more relevant to draw associations in a cancer setting.
The purpose for discussing the relevance of the identified mtDNA mutations in myopathies was to present the known functional/pathological importance of mtDNA mutations in the tRNA region. Not many studies have focused on mitochondrial mutations in the tRNA region and we highlighted some functional/pathological impacts of the identified four mutations of the mt-tRNA region. We have further improved Discussion section by including the relevance of the identified mtDNA mutations in various cancer settings (Revision p.11: Lines 367-368).
Round 2
Reviewer 2 Report
Although the authors did not completely address all my concerns and did not perform the recommended experiments, the authors have added a substantial amount of new data that has strengthened the manuscript. I have no further comments.